# Hazard Analysis and Critical Control Point (HACCP) Application to Dry-Cured Pastrami in Egyptian Pastrami Factories

**DOI:** 10.3390/foods12152927

**Published:** 2023-08-01

**Authors:** Rehab Mohammed El-Mougy, Samir Mohammed Abd-Elghany, Kálmán Imre, Adriana Morar, Viorel Herman, Khalid Ibrahim Sallam

**Affiliations:** 1Department of Food Technology, Faculty of Agriculture, Mansoura University, Mansoura 35516, Egypt; elmougy_2005@hotmail.com; 2Department of Food Hygiene and Control, Faculty of Veterinary Medicine, Mansoura University, Mansoura 35516, Egypt; 3Department of Animal Production and Veterinary Public Health, Faculty of Veterinary Medicine, University of Life Sciences “King Mihai I” from Timișoara, 300645 Timișoara, Romania; adrianamo2001@yahoo.com; 4Department of Infectious Diseases and Preventive Medicine, Faculty of Veterinary Medicine, University of Life Sciences “King Mihai I” from Timişoara, 300645 Timișoara, Romania; viorel.herman@fmvt.ro

**Keywords:** HACCP, dry-cured pastrami, Egyptian pastrami factories

## Abstract

The current study established a HACCP tactic for all hazards related to Egyptian dry-cured pastrami production. All types of hazards that could occur at each production step were depicted. The fabrication steps of pastrami were originally based on the processes and conclusions presented in two previous publications by members of the research team; thus, the current scientific paper is considered a completion of the two previous publications. All operations executed and products manufactured outside the pastrami factory were excluded. The HACCP system was only applied to dry-cured pastrami production stages from receiving frozen raw meat and non-meat ingredients until packaging and storing the final product. Four CCPs were detected and taken into account. The permissible critical limits of additives and non-meat ingredients were considered. Suitable corrective actions were arranged. Regular HACCP plan review, proper recording of detected CCPs and critical limits were necessary for validation processes. Following up with the supply chain by obtaining the certified suppliers, together with the importance of the verification procedures of the elaborated HACCP plan, were essential in order to ensure the production of safe food without defects.

## 1. Introduction

A balanced tactical approach to assessing hazards correlated to food fabrication and the creation of regulation schemes to guarantee food safeness is established and known as HACCP [1]. It is a defensive scheme that deals with the entire process of food fabrication and takes into account the safeness of food goods before biological, chemical and/or physical hazards influence [2]. The evaluation of hazards related to definite food production, identification of manufacturing phases wherever hazards manifest, and outlining of a regular methodology are ensured by HACCP [3].

The HACCP system handles hazards that must be revoked or rescinded to satisfied points and is critical to the fabrication of harmless food [4]. An efficient HACCP tactic should be founded upon a rock-solid basis of prerequisite agendas, good manufacturing practices (GMP) and standard operational procedures (SOP) [5]. Prerequisite programs afford the basic settings that are required for producing safe and high-quality food [6].

GMPs are proceedings applied by food producers to maintain food product safety. GMPs include hygienic settings for staff, fixtures, procedures and ambience in manufacturing progression [7]. SOPs are written depictions of certain responsibilities to be implemented in an exact food treatment operation. They are utilized in conjugation with GMPs to assist a HACCP tactic [8].

Meat and meat products like dry-cured pastrami are deemed to be the food products most vulnerable to being dirtied by pathogenic microbes, as meat represents a rich environment ready for microbial progress. Pastrami is a dried meat product that is cured using salt (sodium nitrite); it goes through several manufacturing stages under special conditions of temperature and relative humidity and may be hazardous for consumers. Since Egyptian pastrami is a popular and desired meat product among Egyptians, researchers and industry officials have had to set rules that ensure the safety of pastrami. 

Therefore, HACCP arrangements in meat fabrication are an imperious and urgent requirement. Some possible microbial threats (e.g., *E. coli O157:H7*, *Salmonella* spp., *Staphylococcus aureus*, *Listeria monocytogenes*, *Campylobacter* spp.) relating to meat products have been demarcated [9,10,11]. These pathogens can be traced to raw meat and meat manufacturing facilities, for example, on tools and machines or in drains. Poor sanitary practices may result in inadequate control. The monitoring of microbial hazards in meat manufacturing or the identification of their occurrence is integrated into the HACCP methodology. Raw meat or meat products may contain some toxic chemical compounds, as raw meat can be contaminated with veterinary drug residues due to illegal practices [12,13].

Therefore, the utilization of veterinary drugs in animals for therapeutic purposes should be applied under strict veterinarian supervision. In addition, drug residue monitoring programs for animals must be implemented in various countries to achieve more surveillance and diminish drug residues in meat. Nitrite salt is widely used in meat production to develop an attractive rose color, but it must be adjusted to permissible amounts in order to avoid carcinogenic impacts [14]. Hence, nitrite salts must be added according to tolerable limits. Other ingredients added to processed meat cause specific allergic impacts. Animal feed and its constituents may contain certain contaminant materials from the environment, like mycotoxins and heavy metals [15]. To avoid economic loss, these dangers must be minimized through HACCP arrangement in combination with GMP in animal rearing and meat manufacturing. 

Hereafter, the conclusions of previous research related to veterinary drugs, preservatives, allergic additives and animal feed containing mycotoxins and heavy metals are exploited to establish a HACCP strategy for all anticipated hazards connected to Egyptian dry-cured pastrami manufacturing procedures, with a special focus on bacterial contamination, aflatoxin issues, veterinarian drug residues and storage problems.

During the fabrication of Egyptian dry-cured pastrami, it may be exposed to poor storage conditions due to a lack of awareness among traders, retailers and even consumers. Meat may be exposed to poor refrigeration and freezing processes and unregulated hygienic practices. In pastrami factories, the temperature and humidity may not be controlled in warehouses with non-meat manufacturing inputs; there may also be a lack of firm programs to control insects and rodents. Even after it has been manufactured, pastrami may be transported from factories to sales outlets during the summer in unequipped cars that do not have temperature control. In retail stores, pastrami may be exposed to high temperatures or insects. Therefore, it is necessary to implement control programs and an effective manufacturing plan to control all sources of danger, not only during manufacturing but also up until the final product reaches the consumer, as well as to educate the consumer on how to handle and store pastrami until consumption.

## 2. Materials and Methods

The defined missions in Codex Alimentarius [16,17] were tracked to create a HACCP plan for all of the predicted hazards (in pastrami manufacturing) that comprise the subsequent seven rules of the HACCP system, namely:

R1: Perform hazard analysis (HA);

R2: Distinguish critical control points (CCP);

R3: Entrench critical limits (CL);

R4: Supervise each CCP;

R5: Reside corrective action;

R6: Demonstrate verification procedures;

R7: Record keeping.

The implementation of the HACCP safety system was based on the passage and development of the successively and complementary stages that were previously mentioned. Thus, the implementation of this proactive system had the main purpose of keeping certain key points in the technological process of dry-cured pastrami in Egyptian pastrami factories under control and demonstrating the fact that the food products resulting in these units are obtained under optimal food safety conditions.

## 3. Results

### 3.1. Establish HACCP Plan and Decision Tree

The stages (related to meat and ingredients production) which operated out of the dry-cured meat manufacturing unit were precluded as the presence of efficient surveillance systems is predictable. These stages applied by suppliers will be recorded via certified documents obtained from suppliers. The HACCP plan will be executed from the reception level of raw materials to the final storage step before distribution. The decision tree diagram (Figure 1) was applied in order to recognize each possible hazard at every phase of production and detect whether the point is a CCP or not [17,18,19]. 

With regard to outlined CCPs, dependable mechanization and corrective actions were settled to achieve the necessities designed by critical limits in order to ensure the safety of products. Verification and archive arrangements were planned to identify the efficacy and tractability of the HACCP tactic.

### 3.2. HACCP Team Formulation

The HACCP team formulation and gathering is considered a momentous issue. Well-trained team members (veterinarians, microbiologists, meat hygiene specialists, etc.) who possess excellent knowledge about the process stages and specialize in implementing each stage are essential to produce high-quality pastrami free from defects. The interdisciplinary HACCP team commanded by the quality supervisor of the processed meat fabrication factory was chosen. The crew adherents were professional veterinarians in food microbiology and meat science specialists who have a wide knowledge of food safety investigation fields, HACCP tactics for a group of food fabrication activities, risk evaluation measures and unscathed food supervision systems.

### 3.3. Product Description

The dry-cured pastrami product is described in Table 1.

### 3.4. Utilized Ingredients in Dry-Cured Pastrami Production

#### 3.4.1. Meat Cuts

Two kilos of *Longissimus dorsi* muscle of frozen beef meat cuts (slaughtered, packed and imported from Brazil; preserved at −18 °C for 2 months) were obtained from a local market in the city of Mansoura, Egypt. 

#### 3.4.2. Non-Meat Additives Used for Curing

The salt (NaCl) was bought from the Mansoura city market by a home-grown Egyptian company. The water exploited for meat rinsing was distilled water used for laboratory tasks. The utilized sodium nitrite (E 250) was manufactured by BASF Aktiengesellschaft in Germany. It was added with an amount of 100 mg/kg meat according to Egyptian standards [21]. The utilized ascorbic acid was bought from El-Gomhuoria Corporation, Mansoura branch. It was used with 550 ppm/1 kg of meat [22]. The sweeteners, including brown sugar and molasses, were obtained from El-Dakahlia Company for Sugar Production.

The spices utilized in curing were nutmeg (*Myristica fragrans*), coriander (*Coriandrum sativum*), clove (*Syzygium aromaticum*), and black pepper (*Piper guineense*). The spices were bought from a reliable supplier (an Egyptian company), which applied the HACCP tactic at every step of production. 

The ingredients other than meat as spices and constituents of seasoning paste were treated via autoclave at 121 °C for 15 min to alter their microbial community and sterilize them as they may be a source of contamination for pastrami during the fabrication process [23]. The autoclave treatment and other kinds of sterilization processes were included in suppliers’ certificates. 

The utilization of HACCP-treated high-quality and autoclaved non-meat constituents enhanced bacteriological quality by minimizing microbial content (total bacterial count and *E. coli O157:H7*) and affirmed manufactured pastrami safeness via destroying *E. coli O157:H7*, which is considered as one of the most essential foodborne microorganisms and diminishing its aflatoxin levels below acceptable aflatoxin border (20 ppb) [23]. 

#### 3.4.3. Non-Meat Additives Used for Seasoning Paste

The seasoning paste was prepared using a grind of roasted fenugreek (*Trigonella foenum graecum*) seeds. The seeds were obtained from a reputable supplier and then roasted and crushed to produce powdery flour, according to Pandey and Awasthi [24]. The resulting fenugreek flour shall be used in pastrami seasoning paste with 500 g/kg meat pursuant to Aksu et al. [25]. The fresh flavored garlic (*Allium sativum* L.) cloves were purchased from a local market in Mansoura. It was cracked. Three percent citric acid was added to smashed garlic for 24 h [26]. The smashed garlic must be kept at −18 °C if it is not used at once.

The fresh sweet (*Capsicum annuum*) and raw hot pepper (*Capsicum baccatum*) were obtained from local markets of Mansoura and utilized in seasoning paste (çemen) applied on pastrami in the form of puree. The puree was prepared by slicing pepper using a knife. The slices were put in hot water at 95 °C for 3–5 min. After that, the pepper was blended in Moulinex Odacio FP7361BM for 5 min to produce a puree [27].

Rosemary and nutmeg were obtained from Mansoura home markets. Ethanolic extracts of rosemary and nutmeg were prepared pursuant to Zamin et al. [28]. 

### 3.5. Altered Pastrami Recipe

The quantities of dry-curing constituents utilized in pastrami manufacturing were inserted pursuant to Aksu et al. [25], in which one ounce of curing combination was used for one meat pound. In the current research, an alteration has been executed by adding the same curing combination along with spices on beef cut to enhance the flavor and quality of dry-cured pastrami and provide an antimicrobial impact on the final product.

For every 1 kg of meat cut, an amount of 62.5 g curing mixture was utilized as follows: table salt (50 g), sodium nitrite (0.1 g), ascorbic acid (0.4 g), brown sugar (2 g), molasses (2 g), coriander (2 g), black pepper (2 g), clove (2 g) and nutmeg (2 g). For the amount (1 kg) of pastrami meat required the formulation of pastrami seasoning cement, the composition was fenugreek flour (500 g), mashed garlic (350 g), sweet red pepper (50 g), hot red pepper (50 g), table salt (20 g), powdered clove (10 g), powdered coriander (10 g), powdered black pepper (10 g), rosemary ethanolic extract (100 mL), nutmeg ethanolic extract (100 mL) and water (800 mL). The dry-cured pastrami constituents are shown in (Table 2).

### 3.6. Pastrami Fabrication

All steps of dry-cured pastrami manufacturing were implemented in accordance with Aksu et al. [25] and Kaban [29], as shown in the flow diagram (Figure 2). The fabrication process is entirely carried out in highly controlled fabrication conditions, as relative humidity should be less than 75% in chambers of drying and pressing, as well as fans which must be utilized to correlate air movement mechanism in fabrication chambers [30]. Also, the temperature in fabrication and meat handling chambers must not exceed 15 °C.

The meat cut was prepared through the deletion of extreme tendons, fat or connective tissues.

Dry-curing: Ten incisions at every side of the meat cut (1 kg) with 1 cm depth and holes of 0.5 cm in diameter were executed in meat pieces to help the curing mixture to penetrate meat depth. Any process of meat handling must be executed in chambers at <15 °C. The curing mixture was distributed to put on the meat surface and penetrate incisions made in the meat. Salted meat was kept at <6 °C for 46 h with a relative humidity lower than 75% (dry curing was applied on the first side for 30 h and then flipped and cured on the other side for 16 h).

First drying: The cured meat cut for pastrami was slightly washed with cold water and then dried at <15 °C in drying chambers with a relative humidity lower than 75% for 4 days in cheesecloth. 

First pressing: The meat cut was pressed using a stainless-steel piston with a pressing power of 5 kg/cm^2^ of the meat surface. Pressing was executed at <7–10 °C for 17 h with a relative humidity lower than 75%.

Second drying: The meat cut was redried for 3 days at <20 °C (with relative humidity lower than 75%) by hanging the meat cut in a cool room supplied with a fan with a velocity of 50 m/s for air renewal. 

Second pressing: It was performed at <25 °C for 7 h with a relative humidity lower than 75% (the same as the first pressing).

Applying of seasoning paste (cement): The meat piece was covered with clay-like seasoning paste (cement). The meat cut was kept with seasoning paste at <7 °C for 4 days with a relative humidity lower than 75%. The excessive layer of seasoning paste (5 mm) was removed, and a thin layer (2–3 mm) of the paste was left on the meat cut.

Final drying: After the application of the seasoning paste, the meat piece was finally dried at 15 °C with 70% RH for 2 days, 18 °C with 65% RH for 2 days, and then 20 °C with 60% RH for 4–5 days to reach water activity 0.9 for pastrami during these days as samples were collected to determine final water activity.

Slicing and packaging: The final pastrami product is sliced via disinfected knives (immersed in water at a temperature of 82 °C or 2% lactic acid for 10 min). Well-trained staff slice pastrami as soon as possible in chambers of a temperature not more than 20 °C. The sliced pastrami final product can be vacuum packed (75% O_2_ and 25% N_2_) and stored in a freezer at −18 °C or in the fridge for 4–5 °C.

### 3.7. Prerequisite Program Application

The implemented experiment in this research illustrated the importance of applying a prerequisite program to stand together with HACCP principles against all kinds of hazards. Hence, we ensured that the application of intact hygienic practices helped produce a wholesome pastrami product starting with sanitation and pest, rodent and insect control during raw material storage up to aeration using fans in order to decrease spore concentration in drying and pressing rooms. As well as personal hygiene commitment, tools and equipment sanitation are also fundamental to getting high-quality products.

Although the manufacturer keeps track of typical good manufacturing practices (GMP), there are some nonconformities related to spore concentrations in manufacturing chambers that may be detected, as demonstrated by Asefa et al. [31], which are briefly discussed below.

The concentration of spores in the air of curing, drying and pressing chambers was identified as an important factor in the potential infection of the pastrami by fungi, specifically by molds, during smoothing. Asefa et al. [31] showed that most of the dry-cured meat foods contaminated with toxic molds were surface-cracked products due to an inappropriate pressing process. The cracks were deemed to be favorable hiding locations and suitable climates for fungi growth and production of their metabolites [31].

Henceforward, it was important to reduce cracks through proper pastrami pressing and drying. Also, the temperature and time of drying and pressing must be carefully adjusted and taken into account. The temperature must not exceed 15 °C. Fans of medium speed (50 m/s) must be used in drying and pressing chambers to circulate air and decrease spore content in the air. Temperature and air circulation monitoring is a fundamental necessity. Additionally, the relative humidity was kept lower than 75% in drying and pressing chambers to avoid mold and yeast growth.

### 3.8. Hazard Analysis, Critical Control Points (CCP), Critical Limits (CL), and Corrective Actions Identified for Every Production Term

In general, the most common hazards found during the manufacturing of Egyptian dry-cured pastrami were biological (microbial and mycotoxigenic), as shown in (Table 3).

#### 3.8.1. Hygienic Condition of Meat at Receiving

Pathogenic yeasts are deemed to be the most common mycological hazards related to raw meat received in the meat production facility. *Candida zeylanoides* was repeatedly detected in the fresh and frozen meat cuts distributed by several abattoirs. The next processing operations, like curing, drying and pressing, postponed the development of the pathogenic yeast and simplified their substitution by non-pathogenic yeast, except for significant types of yeasts such as *Debaryomyces hansenii* [32]. Henceforth, this step did not achieve the standard to be a CCP, so it is deemed to be CP. The very small amounts of harmful fungal metabolites in raw fresh or thawed meat coming from contaminated feeds are the possible significant chemical hazards at this step [33]. 

The danger can only be diminished by “bringing for certificates which demonstrate that the slaughtered animals had experienced a good practice of animal farming, feeding and welfare”. If the animals’ feed is uncontaminated and of sufficient quality, this, in turn, leads to producing high-quality and safe meat [34]. 

Also, the elongation of receiving time of meat cuts is not in favor of the microbial quality of the meat, as the receiving time of frozen meat cuts should not be more than 20 min [35]. The central temperature of frozen meat cuts during receiving must be −12 °C maximum, and it should be measured by a specific thermometer for frozen food under the supervision of a quality engineer at the meat fabrication factory [36].

#### 3.8.2. Thawing

Hazards related to thawing include cross-contamination coming from drip and the progress of microorganisms on the outer part of meat cuts before the inner part has thawed [37]. The meat cuts should be kept in sealed containers, wrappers or protective packages during thawing. The thawing chamber constructed for that purpose is capable of maintaining a temperature that is 4–5 °C in order to guarantee that no portion of the meat reaches a temperature more than 5 °C as the thawing process executed in fridge chambers to treat large meat quantities used in manufacturing field [35]. If the thawing process is executed in water, it must be running in potable water at a temperature not higher than 21 °C for a period of not more than 4 h [37].

#### 3.8.3. Trimming

The removal of bones, tendons and connective tissues is the applied activity in this stage of production. The undesired yeasts as *C. zeylanoides* are still one of the most anticipated hazards that may present, according to Asefa et al. [32]. The trimming procedure can also be a source of another biological hazard if the meat cuts are left in a high-temperature chamber for a long time. So, it must be applied under suitable hygienic conditions, including equipment and utensils. The temperature of trimming chambers must be ≤10 °C, and meat pH must be 5.5 to 5.8 [38]. The time of meat trimming or preparation must be reduced as much as possible; typically, 30 min or lower. In large-scale production, the temperature of trimming areas must not exceed 15 °C while keeping the temperature at 4 °C inside the core of the meat cuts with regular monitoring of the core temperature [37].

#### 3.8.4. Dry-Curing of Meat Cuts

After trimming the meat cuts, the dry curing mixture shall be put on the meat surface and properly rubbed. Salted meat was kept at 6 °C for (30 h on one side and 18 h on the other side) [25], and RH was lower than 75% [30]. To avoid the anticipated hazards in this stage of production, services and incisions must be made in meat cuts using an aseptic knife to facilitate good salt penetration with proper monitoring of temperature and humidity along the stage. The dry-cured meat cuts must be put in stretch plastic filaments or polyethylene bags vacuumed from the air to reduce microbial growth.

#### 3.8.5. The First Drying and First Pressing of Meat Cuts

These two stages of production are deemed to be CCPs, as drying and pressing in the absence of a curing mixture can raise the microbial content. In first curing, the dry-cured meat cuts are slightly rinsed with water (60–65 °C) [38]. The water rinsing process must be rapid with minimal water quantity to avoid making the pastrami product wetter and increasing its water activity. The high water activity of dry-cured products leads to facilitating the proliferation of toxigenic molds and might lead to the potential production of toxic secondary metabolites in the products [30]. The research team proposed to dry meat cuts with clean food-grade gauze to remove any water residues from rinsing.

Afterward, the meat cuts are dried at <15 °C for 4 days [25] (Aksu, 2017), and relative humidity should be adjusted to lower than 75% (except for the washing step). Also, fans (50 m/s) must be utilized to improve the air movement system in the drying and pressing chambers [30]. The drying and pressing rooms must be 12–15 °C [30,37]. In the first pressing, the meat cut was subjected to pressing via a sterilized stainless-steel piston at <7–10 °C for 17 h, also with RH lower than 75% [25]. To avoid the anticipated hazards in two such stages of production, the temperature and humidity must be properly monitored, using potable autoclaved water in meat rinsing, put the hygienic condition of pressing piston surface (the surface attached to meat cuts) into account and fans must be utilized to improve air movement system to circulate air and decrease spore content in the air with assuring that the fans and piston are maintained periodically. 

#### 3.8.6. The Second Drying and Second Pressing

These two stages of production are deemed to be CCPs due to the elevation of stage temperature up to <25 °C. In the second drying, the meat cut was redried for 3 days in an environment of <20 °C with a relative humidity lower than 75% [25]. The second pressing was performed at <25 °C for 7 h via a stainless-steel piston. To avoid the anticipated hazards in two such stages of production, proper monitoring of temperature and humidity must be applied, and hygienic condition of the pressing piston surface must be taken into account, as well as fans (with a velocity of 50 m/s), which must be utilized to improve the air movement system to circulate air and decrease spore content in the air in the drying and pressing chambers with proper maintenance of the piston and fans. 

It is important to check the water activity at the end of the drying and pressing stages, as the ability to produce toxic secondary metabolites of *Verrucosidin* and *Penicillium polonicum* insulated from dry-cured ham is highly affected by the water activity of the product [39]. A drying process that lowers the water activity of the products faster to a level near 0.9 minimizes the risk of toxic secondary metabolite production tremendously. Mycotoxins production by toxigenic molds was not observed at a water activity of 0.9 [40]. For this reason, the research team decided to adjust the critical limits of water activity at the end of the first pressing stage to 0.92 and at the end of the second pressing to 0.9.

#### 3.8.7. Addition of Seasoning Paste

This step is considered a CCP as the seasoning paste is composed of many constituents, which must be aseptically prepared and applied on the meat surface, taking into account good hygienic conditions. It can elevate the microbial count and pathogenic molds, which can exist in cracks formed at the surface of the seasoning paste.

In order to avoid the anticipated hazards in such a stage of production, the time of application of seasoning paste on meat cuts must be 30 min or lower in chambers with temperatures which must not exceed 15 °C [37]. Also, the seasoning paste must be slurry enough in order to avoid making cracks. Meat cuts should be kept with seasoning paste at <7 °C for 4 days with relative humidity lower than 75%. The extreme layer of cement would be detached, and a thin layer (2–3 mm) of the paste would be left on the meat cuts [25]. Also, continuous supervision should be given for crack formations on the surface of the meat cuts.

#### 3.8.8. Final Drying

This stage of production is considered a CCP due to the gradual increase of temperature up to <20 °C and the probability of forming cracks at the surface of a thin seasoning layer which is deemed to be a good environment for yeast and molds growth. Also, the meat cut was hung through strong threads for drying in specific chambers of elaborate temperature. After the seasoning paste application, the meat cut was subjected to the final drying in an atmosphere of <15 °C with 70% RH for 2 days, then at <18 °C with 65% RH for 2 days, and finally at <20 °C with 60% RH for 4–5 days [25]. The temperature and humidity must be cautiously monitored in order to avoid the expected hazards at such a stage of production as well as the water activity should not be more than 0.9. The CCPs were identified in the flow diagram of production stages, as shown in Figure 3. 

### 3.9. Final Product Slicing and Storage

The final dry-cured pastrami product was sliced after the removal of the seasoning paste layer and packaged in vacuumed polyethylene bags (75% O_2_ and 25% N_2_), taking into account the rigorous personnel hygienic conditions. The final product was kept in frozen storage (−18 °C) or at 4 °C in the fridge for up to 60 days [23].

### 3.10. Residing of Corrective Actions

The corrective actions are measures implemented in order to correlate any deviation that may happen throughout the fabrication steps from the permissible critical limits. These limits were established via the public authorities and other international organizations [2]. The suitable corrective actions taken by the research team were demonstrated in Table 3 and Table 4.

### 3.11. Verification Process and Record Keeping

The periodic visual observation by officials in meat plants is deemed to be one of the most beneficiary steps in executing verification. Samples of pastrami can be taken at various stages of production to check water activity, total viable count and toxigenic molds. Temperature, water activity, duration of each production step and mold and yeast count are deemed to be some critical aspects of dry-cured pastrami manufacturing. The process of device calibration is extremely important, especially for devices measuring water activity, temperature, relative humidity, fans’ velocity, piston pressure and any other important device used along the manufacturing process to keep their records. 

### 3.12. Validation

For obtaining high-quality HACCP tactics, it is vital to formalize the inspection and visitation processes by following up on the legal regulations at local and international levels with official authorities. The initial validation and recurring revalidation of HACCP plans must be included in such regulations. Mandatory verification of awareness and skillfulness of specialists providing consultation services related to food safety is originally required [41]. 

The whole HACCP tactic reviewing, proper marking of records, identifying the permissible critical limits, enumeration of deviation cases, execution of corrective actions, product sampling and analysis at every stage of production with maintenance and calibration of all utilized devices were considered to be effectual in implementing validation processes within a meat plant [2,42,43].

## 4. Discussion

The main object of such an industrial experiment is to establish a HACCP plan for dry-cured pastrami to ensure there are no defects and the product is safe. Therefore, the consumption of high-quality products can eliminate foodborne illness. The research concluded that there are four CCPs during the pastrami manufacturing process, and they are closely related to the temperature, humidity and time of product drying and pressing, which is reflected in the product water activity at the end of each production stage. This required controlling and calibrating temperature and humidity devices regularly. It also required the usage of appropriate ventilation tools in the drying and pressing rooms to renew the air to avoid gathering spores on pastrami. 

In addition to identifying CCPs, the application of the HACCP system led to the commitment to the additives’ rates and product components, examining the quality of non-meat additives involved in manufacturing, and following their production and supply chain until they reach the factory, which is consistent with the desired goal of that research paper.

These results are also in line with the findings of researchers in the field of food safety and food industries, with Asefa et al. [30] concluding that conditions and activities that maintain good hygiene and reduce fungal contamination in cured meat products are necessary. Metaxopoulos et al. [10] also showed that the HACCP system and GHP were applied in Greece via the meat industry and proved the importance of safety assurance programs’ application together with high-quality raw materials, product handling and staff training. Additionally, Hasnan and Ramli [44] demonstrated that the HACCP system allows suitable handling activities, hygienic conditions and effective safety procedures, which presents valuable knowledge for entrepreneurs. Furthermore, Wang et al. [45] showed that the effective HACCP system and plan can eliminate chemical and microbiological contaminants in vacuum-packed sauced pork.

This study is considered comprehensive and deals with all points of manufacturing pastrami products, which can be generalized among many states who prefer this type of processed meat, taking into account temperature, humidity and time of processing with constant calibration of measuring devices, adjusting meat receiving and storage practices, and training workers [11] on periodic monitoring and knowing the severity of each CCP. This demonstrates the possibility of expanding the production of dry-cured meat products while adhering to quality specifications, correct ratios of additives, good manufacturing practices and proper storage.

With the difference in the common sense of consumers in each country, the additives or the manufacturing methods may differ; this requires more research to be conducted on the manufactured product to examine the presence of new CCPs that may lead to health risks to consumers.

Odintsova and Panin [46] were striving to generate a quality system for the manufacturing process of canned meat products utilized in nourishment. In this paper, they settled the manufacturing process of canned meat products, described the CCPs, and created a HACCP plan. Through these collected data, they could execute a HACCP system. There were three CCPs along with the technological process, which could cause risks for children as target consumers. It was concluded that the HACCP application was an effective tool to avoid such health risks.

Poumeyrol et al. [47] established a methodology for meat pâté as a practical example to identify and assess the bacterial hazards, which led to discrimination between hazards that can be controlled via PRPs and those that can be monitored by GHP and GMP. In addition, the hazard analysis of meat pâté fabrication revealed three possible hazards that appeared in three CCPs, control of *C. botulinum*, *C. perfringens* and *B. cereus*, respectively. Application of suitable corrective actions and effective hygienic practices within a firm HACCP plan could eliminate the microbiological profile of meat pâté final product.

Henriques et al. [48] performed an analysis of meat product samples and auditing in Portuguese industrial factories to determine the level of hygienic practices of staff and manufacturing practices via a checklist. They concluded that the unacceptably high levels of *Listeria monocytogenes* in ready-to-eat meat products were closely related to poor hygienic conditions and practices. In other words, the application of a strict hygiene program and a correct HACCP plan is one of the most important reasons that help in controlling biological factors in the final product.

Subramaniam et al. [49] evaluated the danger of microbial contamination in ready-to-eat poultry products in a number of HACCP-certified food manufacturing establishments in Malaysia. The results were alarming due to the presence of *S. aureus* and *coliforms* in poultry-ready products. Additionally, the contaminated contact surfaces and staff hands were detected, which revealed the importance of more firm HACCP plan application in these facilities with continuous internal and external auditing to minimize foodborne pathogens to permissible limits.

Yang et al. [50] performed a study introducing a meta-analysis to clarify the relationship between HACCP-based FSM system application in small and/or less developed food businesses (SLDBs) in China and their food safety conditions. The results ensured the importance of HACCP-based FSM system application in Chinese SLDBs with variable products, business scale, employees’ education and culture, training level and employees’ understanding of food safety concepts.

Altogether, the results presented in the current investigation underlined the fact that following up with the supply chain through obtaining and selecting certified suppliers is an extremely essential proceeding in order to ensure the production of safe and nutritious foods without defects.

## 5. Conclusions

This research paper affirms the importance of the application of the HACCP tactic in food plants, especially pastrami factories, because meat is one of the most vulnerable food types that can develop pathogenic microbes. HACCP came as a saving tool in pastrami fabrication stages to ensure products had zero defects. The research team indicated that all aspects of the HACCP tactic in pastrami manufacturing, including the prerequisite programs, were fundamental aspects to achieving HACCP plan efficacy. Monitoring and correction of deviation were revealed as basic pillars in maintaining permissible limits. The four logically identified CCPs in pastrami manufacturing were correlated to microbial hazards. The CCPs monitoring was mainly conjugated with critical limits of water activity. Also, adjustment and monitoring of temperature and every stage duration are mandatory, with keeping all critical documents related to microbial counts and toxigenic metabolites at each production step.

The staff’s commitment to good practices towards all manufacturing steps came from their awareness of the importance and significance of every step, including CCPs. Good training of employees will achieve this point and raise their awareness and knowledge.

## Figures and Tables

**Figure 1 foods-12-02927-f001:**
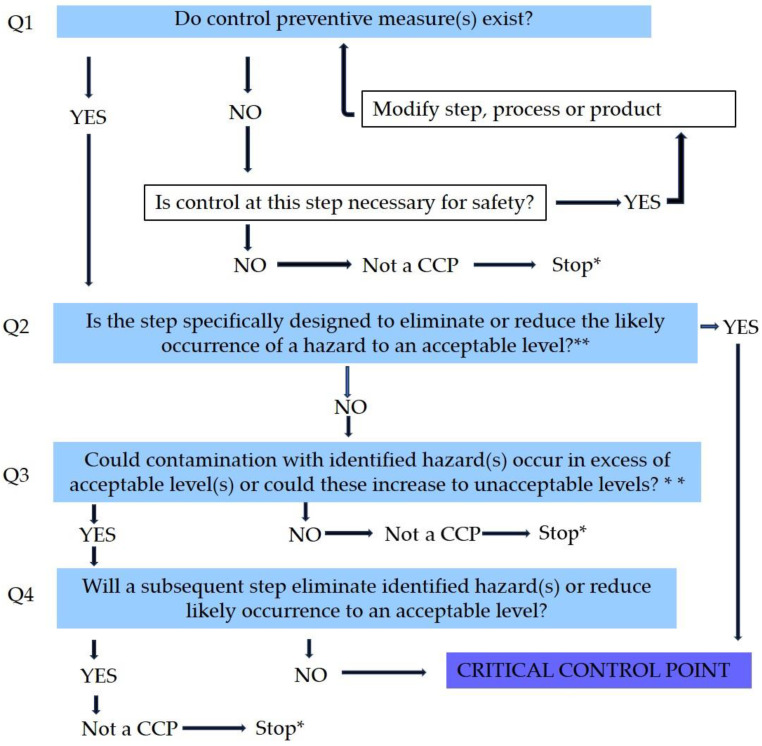
The decision tree diagram for critical control points (CCPs). Legend: * proceed to the next identified hazard in the described process; ** acceptable and unacceptable levels need to be determined within the overall objectives in identifying the CCPs of the HACCP plan [18].

**Figure 2 foods-12-02927-f002:**
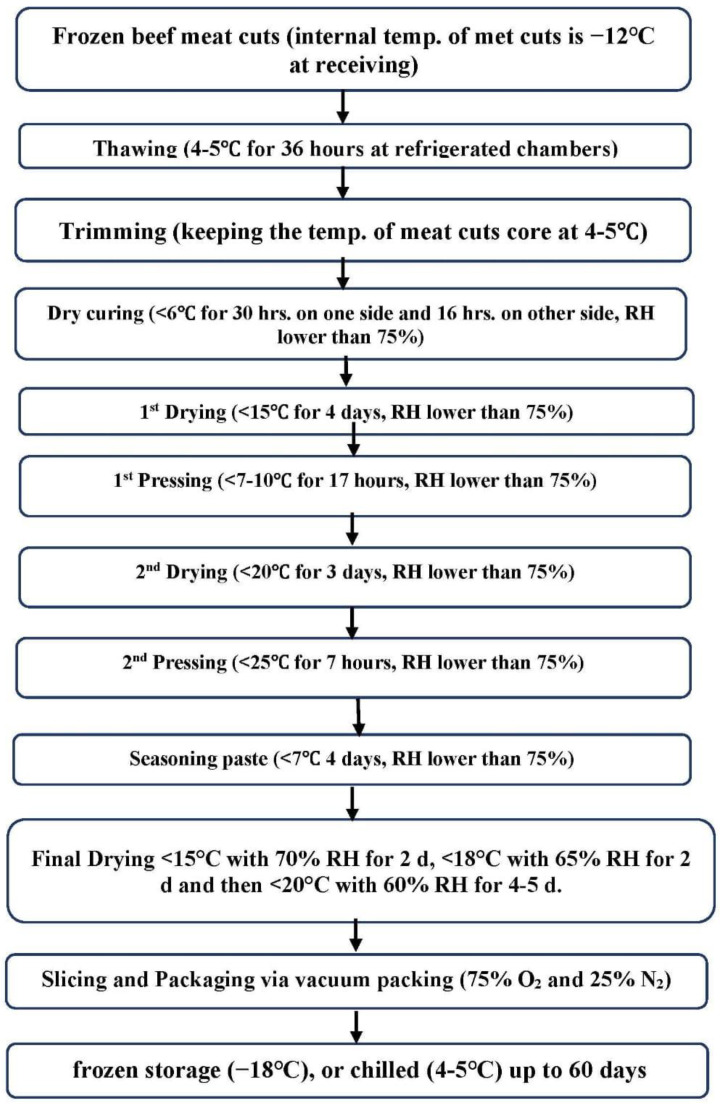
Flow diagram of the dry-cured pastrami production [25,29].

**Figure 3 foods-12-02927-f003:**
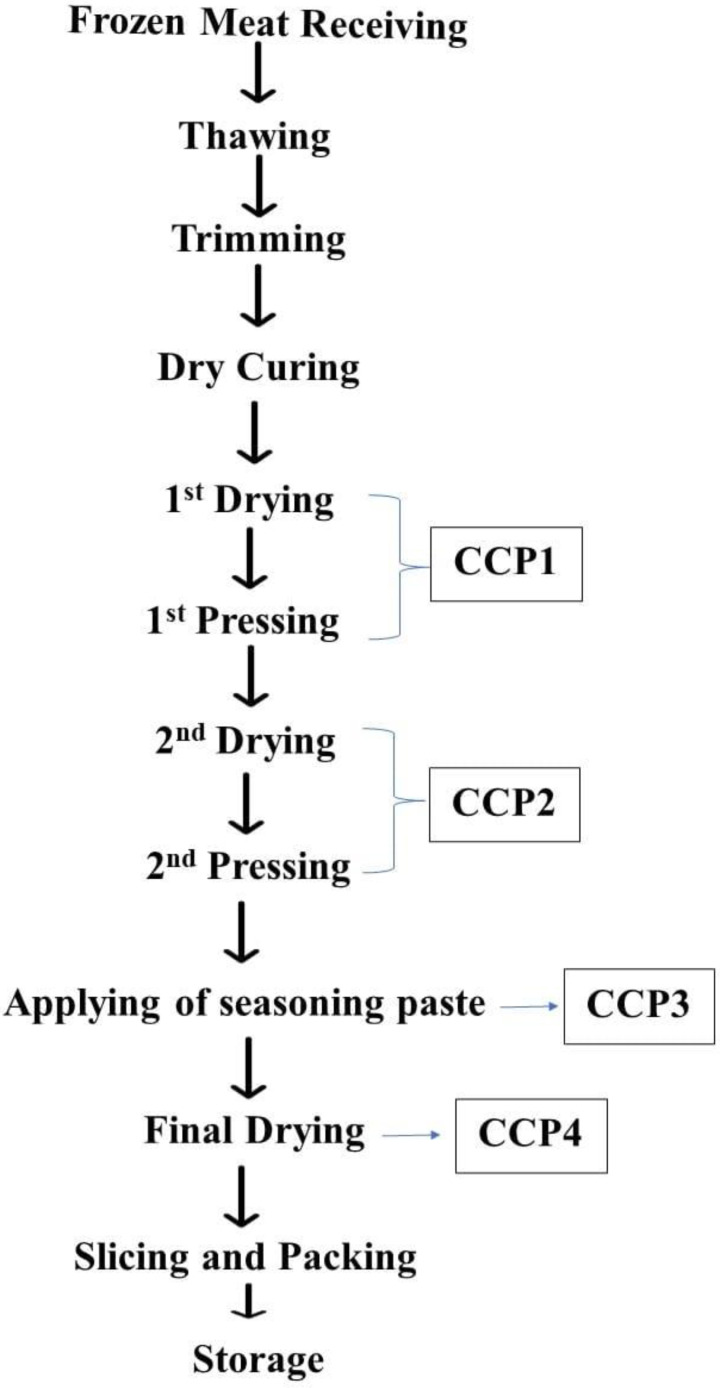
CCPs within the dry-cured pastrami flow diagram.

**Table 1 foods-12-02927-t001:** Product description of dry-cured pastrami.

Product Name	Egyptian Dry-Cured Pastrami
Meat cut	*Longissimus thoracis* et *lumborum* (*Longissimus dorsi muscle*)
Additives (curing mixture)	Table salt (7%); sodium nitrite (100 ppm), ascorbic acid (550 ppm); spices; brown sugar; molasses
Seasoning paste constituents	Fenugreek; garlic; hot and sweet pepper puree; rosemary and nutmeg extract; spices; water for mixing
Processing method	Dry-curing mixture application: drying, pressing and covering with seasoning paste; final drying
Product characteristics	Sensory index: the product has a natural color, smell and taste, no foreign body attached, and no bad odor. Sanitary index: water activity (0.9), pH (6.0). Aerobic plate counts 10^2^ CFU/g, absence of *E. coli O157:H7*
Packaging Usage	Vacuumed polyethylene bags Ready-to-eat (or cooked as desired)
Storage temperature	Cold storage via freezing at −18 °C (in the freezer) or at 4 °C (in the fridge) [20]
Storage period	60 days from the production date [20].

**Table 2 foods-12-02927-t002:** Egyptian dry-cured pastrami ingredients.

Parameter	Level
Water Activity (a_w_)	0.9
Salt	7%
NaNO3 (E250)	0.01%
Protein	29%
Fat Lean Meat	5% 95%
Ascorbic Acid (Antioxidant)	0.04%
pH	6.0

**Table 3 foods-12-02927-t003:** Critical control points and corrective actions in dry-cured pastrami production.

Ingredient/ Process of Manufacture	Activity	Nature of Anticipated Hazard	Hazard	Is It a CCP? If Yes/Risk of Occurrence	Applied Control Measures/Monitoring to Prohibit the Anticipated Considerable Hazards	CCP
Receiving Raw Meat	Receiving frozen meat. Meat may be a great source of microbes if badly handled, packed or stored.	Biological	Bacteria: *E. coli O157:H7*, *Salmonella* spp., *S. aureus*, *L. monocytogenes*, *Campylobacter* spp., *Y. enterocolitica*, *Cl. botulinum*, *Cl. perfringens*, *Cl. zeylanoides.* Parasites: *Trichinella* spp., *C*. *cellulosae*, *C. bovis*, *T. gondii*, *Taenia solium.* Viruses: hepatitis E virus, norovirus, rotavirus, astrovirus. Yeasts: *Candida zeylanoides*.	Yes/ High	Food safety programs in animal farms and abattoirs must be applied with the availability of reliable documents, such as the guarantee letter from the supplier, to ensure that meat is produced according to hygienic and governmental standards (good feeding and welfare). Meat proper freeze at −18 °C to control pathogens, and meat cuts shall be received within 20 min via specialists as the temperature of inner meat cuts should not be more than −12 °C. Adjusting and monitoring the freezing temperature. The control occurs through subsequent steps.	CP (GMP)
		Chemical	Antibiotic/drug residues: hormones, pesticides, heavy metals, dioxins. Biogenic amines: mycotoxins.			
Meat Thawing	The frozen meat shall be defrosted.	Biological	Microbial content elevation of meat.	Yes/ Medium	Meat thawing in the fridge at 4–5 °C. Proper temperature adjustment to avoid increased microbial load. More control shall be achieved in the subsequent steps.	CP (GMP)
Meat Trimming	Deboning of defrosted meat cuts and excessive CT removal.	Biological	It can be a remarkable source of pathogenic yeasts and bacteria.	Yes/ Medium	GMP shall be applied using gloves, masks and personal hygiene. The knives used must be disinfected via immersion in water at a temperature of 82 °C or in 2% lactic acid for 10 min. Well-trained workers should trim meat cuts as quickly as possible to maintain the internal meat temperature stay between 4–5 °C. Yeast and mold growth will be controlled by subsequent steps. More control shall be achieved in the subsequent steps.	CP (GMP)
Dry-curing Step	Dry curing mixture addition and keeping meat cuts at 6 °C for 46 h with RH lower than 75%.	Biological	It can be a root of microbes (as added spices may be a source of meat cuts contamination) or insufficient permeation of salt into meat cuts.	Yes/ Medium	Cervices and incisions must be made in meat cuts using an aseptic knife (immersed in water at a temperature of 82 °C or in 2% lactic acid for 10 min) to facilitate good salt penetration. Also, the added spices must be obtained from an authorized supplier with clarification of sterilization processes performed for spices in quality certificates. The meat cuts must be kept at 6 °C for 46 h, RH lower than 75%, with proper monitoring of temperature and humidity along the stage. The dry-cured meat cuts must be put in stretch plastic filaments or polyethylene bags vacuumed from the air to reduce microbial growth.	CP (GMP)
1st Drying and 1st Pressing	1st drying implemented at <15 °C (4 days), and 1st pressing at <7–10 °C for 17 h with RH lower than 75%	Biological	It can elevate the microbial count.	Yes/ High	Temperature and time of drying and pressing must be carefully adjusted and taken into account. The temperature must not exceed <15 °C with monitoring RH lower than 75% with getting rid of meat drip regularly every 4 h. Fans of medium speed (50 m/s) must be used in drying and pressing chambers to circulate air and decrease spore content in the air. Temperature and air circulation monitoring is a fundamental necessity. Water activity must be monitored at the end of such stage of production and must be at 0.92 at the end of the 1st pressing.	CCP1
2nd Drying and 2nd Pressing	2nd drying implemented at <20 °C (3 days) and the 2nd pressing at <25 °C for 7 h with RH lower than 75%.	Biological	It can elevate the microbial count.	Yes/ High	Temperature and time of drying and pressing must be carefully adjusted and taken into respect. The temperature must not exceed <15 °C with getting rid of meat drip regularly every 4 h. Fans of medium speed (50 m/s) must be used in drying and pressing chambers to circulate air and decrease spore content in the air. Temperature, humidity, and air circulation monitoring is a fundamental accomplishment. Water activity must be monitored at the end of such stage of production and must be at 0.9 at the end of 2nd pressing.	CCP2
Seasoning Paste Preparation and Application	Aseptic and hygienically treated contents of seasoning paste are thoroughly mixed and applied on meat cuts at <7 °C for 4 days and RH lower than 75%.	Biological	It can elevate the microbial count and pathogenic molds, which can exist in cracks formed at the surface of the seasoning paste.	Yes/ High	A clean and sanitized container and spatula are used to mix the seasoning paste content to be a homogeneous and slurry mass. The time of application must be controlled and not exceed 30 min. Meat cuts are covered with seasoning paste (slurry enough to avoid cracks formation) by gloved hands. The dryness of seasoning paste must be avoided to prevent crack formation. Meat cuts were kept with seasoning paste at <7 °C for 4 days and RH lower than 75%. GMP, sanitation, and personal hygiene must be applied.	CCP3 (GMP)
Final Drying	The meat cuts covered with paste would be dried again at <15 °C, 70% RH for 2 days, at <18 °C, 65% RH for 2 days, and then in an atmosphere of <20 °C, 60% RH for 4–5 days.	Biological	It can elevate the microbial count.	Yes/ High	The temperature, duration and humidity must be cautiously monitored. Hanging of dry-cured pastrami for final drying must be executed under complete hygienic conditions with periodical observation of cracks formation in the layer of seasoning paste. The cracks must not be formed in seasoning paste-covered pastrami. Any formed cracks should be plugged immediately with a seasoning paste called çement to avoid mold growth in such cracks. Samples of pastrami must be collected in the last 4–5 days to ensure that water activity reaches 0.9	CCP4
Slicing and Packaging	The pastrami final product can be vacuum packed (75% O_2_ and 25% N_2_) and kept in the fridge (4–5 °C), or it may be sliced with the removal of paste, vacuum-packed, and frozen at −18 °C.	Biological	It can be a root of bacteria and fungi.	Yes/ Low	The GMP shall be applied during handling and slicing using aseptic knives. Storage temperature must be well-adjusted and monitored.	CP (GMP)

**Table 4 foods-12-02927-t004:** Critical control points and corrective actions in dry-cured pastrami production.

CCPs	Hazard Description and HACCP Plan				
	Critical Limits	Monitoring	Corrective Actions	Verification	HACCP Records
CCP1 Control microbial content of pastrami (total viable count TVC and pathogenic yeasts) 1st drying and 1st pressing	<15 °C, 4 days (1st drying). <7–10 °C for 17 h with RH lower than 75% (1st pressing). At the end of 2 such stages, a_w_ must be 0.92	▪ Monitoring of temp./duration/ RH/a_w_ control. ▪ Changing the gauze every 4 h. ▪ Monitoring of performance of pressing piston. ▪ Monitoring of drip and removing it every 4 h. ▪ Ensure that the fans are running efficiently at 50 m/s.	▪ Correct temp. and duration ▪ Calibration of temp. probes and adjusting RH. ▪ Using fans to circulate air in drying and pressing chambers to avoid the accumulation of spores. ▪ Ensure that the fans are maintained periodically.	▪ Temp. and RH check. ▪ a_w_ measure at the end of 2nd pressing. ▪ Quality control audit and calibration of utilized apparatuses	▪ Performance records (Temp./duration/RH/a_w_ control)
CCP2 Control microbial content of pastrami (TVC and pathogenic yeasts) 2nd drying and 2nd pressing.	<20 °C, 3 days (2nd drying). <25 °C for 7 h at RH lower than 75%. At the end of 2 such stages, a_w_ must be 0.9.	▪ Monitoring of temp./duration/ RH/a_w_ control. ▪ Monitoring of utilized gauze and changing it every 4 h. ▪ Monitoring of the performance of pressing piston ▪ Monitoring of drip and removing it every 4 h. ▪ Ensure that the fans are running efficiently at 50 m/s.	▪ Correct temp. and duration ▪ Calibration of temp. probes and adjusting RH. ▪ Using fans to circulate air in drying and pressing chambers to avoid the accumulation of spores. ▪ Ensure that the fans are maintained periodically.	▪ Temp. and RH check. ▪ a_w_ measure at the end of 2nd pressing. ▪ Quality control audit and calibration of utilized apparatuses.	▪ Performance records (Temp./duration/RH/a_w_ control).
CCP3 Control microbial content of pastrami and seasoning paste (TVC and pathogenic yeasts)	Meat cuts were kept with seasoning paste at <7 °C for 4 days at RH lower than 75%.	▪ Monitoring of temp./duration/ RH/a_w_ control. ▪ Slurry and smooth paste must be supervised to avoid cracks formation.	▪ Correct temp. and duration. ▪ Calibration of temp. probes and adjusting RH. ▪ Using fans to circulate air in drying and pressing chambers to avoid the accumulation of spores. ▪ The paste must be slurry enough to avoid cracks formation.	▪ Temp. and RH check. ▪ Quality control audit and calibration of utilized devices.	▪ Performance records (Temp./duration/RH/a_w_ control).
CCP4 Control microbial content of pastrami (TVC and pathogenic yeasts) in the final drying stage	<15 °C with 70% RH for 2 days, <18 °C with 65% RH for 2 days and then <20 °C with 60% RH for 4–5 days	▪ Monitoring of temp./duration/ RH control. ▪ Monitoring any cracks at the layer of seasoning paste which covered the product. ▪ Ensure that the fans are running efficiently at 50 m/s.	▪ Correct temp. and duration. ▪ Calibration of temp. probes and adjusting RH. ▪ Using fans to circulate air in drying and pressing chambers to avoid the accumulation of spores. ▪ Seal any appeared cracks. ▪ Ensure that the fans are maintained periodically.	▪ Temp. and RH check. ▪ Quality control audit and calibration of utilized devices.	▪ Performance records of temp., duration and RH

## Data Availability

All data are included in the present manuscript.

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
