# Peer review of "Hazard Analysis and Critical Control Point (HACCP) Application to Dry-Cured Pastrami in Egyptian Pastrami Factories"

_foods, 2023, doi:10.3390/foods12152927_

Round 1

Reviewer 1 Report (Previous Reviewer 3)

The manuscript is presenting and described Hazard Analysis and Critical Control Point (HACCP) application to dry-cured pastrami in Egyptian pastrami Factories.

In the reviewer's opinion, the manuscript submitted for review is more of an application than a scientific nature. I have pointed this out in my previous reviews. The resubmitted version contains all the classic sections that must be included in a scientific article, namely: "Introduction", "Materials and Methods", "Results", "Discussion", "References", which does not make it after all a classic scientific paper.

In fact, the paper is a description of the procedure introduction ensuring food and consumer health safety, dedicated to a specific industrial plant. An important drawback of the manuscript is still the lack of clearly and legibly indicated the aim of study. Nevertheless, despite the indicated shortcoming, the content of the reviewed manuscript is interesting and valuable and may be useful to meat processing practitioners.

After carefully reviewing the substantive value of the manuscript, I believe that improved version the article can be published in Foods.

I leave the final decision on qualifying the article for publication to the Editor-in-Chief of the section.

Author Response

Dear Reviewer,

Once again, special thanks for your efforts in reviewing our mansucript and your valuable comments which have greatly contribute to increase its quality. We hope that our efforts will be positively appreciated by the Editor, and the present version of the manuscript will be considered for publication in the prestigious Foods journal. Moreover, addressing to the raised concerns by another two reviewers in this review round have brought substantial improvements to the manuscript.

Thank you again!

Reviewer 2 Report (Previous Reviewer 2)

Despite the discussion having been reformulated, it remains deficient. Several results were presented in tables and figures, but little discussed, with few paragraphs and without technical relevance when compared with other papers. Overall, the article is poorly distributed in this context. There are only five discussion paragraphs. Also, the conclusion is not well written either, citing tables in the description. Few important changes were made from the previous version to this one and it remains with relevant tweaks to be corrected.

Author Response

Dear Reviewer,

Once again, special thanks for your efforts in reviewing our mansucript and your valuable comments which have greatly contribute to increase its quality. According to your suggestion, the revised version of the manuscript contain the necessery improvements requested for the Discussion section, meaning the insertion of several new paragraphs (Please see the lines 462 – 499 of the revised version). In addition, the conclusion section has been substantially revised addressing to the raised concerns by the reviewer (Please see the lines 501 – 520 of the revised version).

Thank you again!

Reviewer 3 Report (New Reviewer)

Overall, the manuscript contains very important information for producing a safe and quality dry-cured pastrami. There are details that need to be taken into consideration regarding the CCPs in the HACCP plan.

Abstract: The authors have not mentioned the importance of verification in a HACCP plan. Could consider incorporating verification procedures in sentence  on line 29 and 30.

Lines 45-46: Prerequisite programs in quality and safety programs are the foundation of safety programs and do not result in secure and healthy food. Authors may want to consider changing to safe and quality food.  Company could have prerequisite programs for a candy bar but that wouldn't be considered healthy.

Lines 55-59: Change to Pastrami is a dried meat product that is curing using salt, sodium nitrite, it....for consumers.

Line 60: Consider starting as a new paragraph.

Line 61: Consider changing to E. coli O157:H7, spell out Staphylococcus

Line 63: Change to These pathogens can be traced to the raw meat and at meat manufacturing facilities on tools and machines or in drains.

Line 64: Consider changing from bad to inadequate control

Line 67: Consider changing from dirtied to contaminated

Line 72: Change to: The preservative use of nitrite salt...

Line 85: Go into a little more detail to explain where the storage problems occur such as at the meat manufacture, retail store, or by the consumer.

Line 110: Change term to phase or step of production

Table 1: For Product characteristics is "absence of E. coli" mean generic or O157:H7?

For the Storage period it looks like it was repeated in the Table. Also, do the temperatures mean the temperature of the storage room or the temperature of the product? Is there a defined or recommend shelf-life for the frozen product?

Line 148: Explain what is meant by autoclave non-meat constituents? Either define the non-meat constituents or why the autoclave process. Is this something that can be done by various suppliers?

Line 149: Do the authors mean generic E. coli or STEC E. coli?

Line 159: Change to city and Three percent citric acid.

Line 161: Consider changing to Fresh or raw sweet ...and hot pepper.

Line 166: Change to city

Line 175: Change to salt

Line 192: Explain approximately how many incision per an area...also how deep are the incisions and what is the diameter of the holes? 

Line 194: Is the 6C the temperature of the refrigerator/drying chamber or the internal temperature of the meat? If temperature of room then consider adding a range such as 6C +/-1C.

Line 195: Is 15C temperature of the drying chamber? Again, if drying chamber temperature again provide a range or consider <15C. What was the humidity for the first drying? Consider adding washed with cold water

Line 198 What was the humidity of the room?

Line 199: The range of 3 to 4 days is pretty wide. Consider adding in the water activity that was trying to be achieved. Also, consider including a range for 20C. Was the meat cut in the cool room prior to the drying step? If so what is the temperature of the cool room? Also what was the humidity? Did the authors measure the velocity of the fan for air renewal?

Line 201: If 25C for drying chamber, should it be <25C or should it be a range? What was the humidity?

Line 203-204 Should it be cement or cemen? Check document for use of the term. Should there be a range or for 7C? Also, 4-5 days is a wide range so should the target water activity be added?

Line 205: Define what is meant by excessive layer such as so many mm as for the thin layer.

Figure 2. The dry curing step is for 46 hours and does not match the time of 48 hours  on Line 194 and Line 194 doesn't mention dry curing on one side and then flipping and drying on the other side.

For Figure 2 and Lines 209 to 211 the authors should consider a range for the 15C, 70% RH, 18C, 65 RH, 20C and 60 RH. For the 4-5 days there should be the target water activity to explain the range of 4-5 days. 

For the Figure 2 and Lines 192 to 211, the authors should include if there is fan control and the speed or velocity during those processing stages.

Line 217: Change to up to 

Line 218: Change to hygiene

Line 219: Change to fundamental

Line 222: Change were to are

Line 232: Can the authors define the velocity of medium speed?

Line 243: Consider changing from repeatedly segregated to repeatedly detected.

Table 3: For the Hazards make sure to spell out all of the names of the pathogens. Also, are you meaning generic E. coli or STEC E. coli? In the receiving of meat is it the internal temperature of meat that should be at -18C or the freezer temperature. In Figure 2 authors have meat cuts (-18C) leading reader to think the meat cut temperature should be -18C at receiving. Need to clarify.

Table 3: Meat thawing should it be Meat thawing with room temperature at 4-5C? 

Table 3: The control mentions knives must be disinfected. Provide details of how this could be done such as hot water. For the meat trimming, the control indicates that meat cuts should be at 4-5C. The sentence could be improved...

Well-trained workers should trim meat cuts as quick as possible to maintain the internal meat temperature stay between 4-5C.

For the Dry-curing step again it is different than in Figure 2 also address 6C and 48 hours and Relative humidity changes as discussed above. For Biological: Should it be root growth of microorganisms? For control, suggest way to obtain an aseptic knife especially in a factory setting.

For the remaining stages address the ranges or less than or greater than for the temperatures, times, RH and fan speed. 

Season paste control: it would be better to indicate a clean and sanitized container and spatula are used. Sterile is not feasible in food industry.

For the Final drying is there a way to define the size of cracks or should there not be any cracks in the layer of seasoning paste. Here it is called paste is this the same as cement?

Slicing and packaging step is not included in the text after final drying or in Figure 2. Explain the parameters for vacuum packaging such as the percent of oxygen. 

Table 4: Should be the first time using TVC so spell out as total viable count.

Again for the critical limits revisit to provide ranges or less than or greater than. This is very important in a HACCP plan especially from a monitoring and regulatory point. If the temperature in the room is only one number then it can be out of compliance if lower or higher. Also, this could increase the number of corrective actions taken.

For the monitoring section would suggest including what specifically is being monitored such as the drying chamber temperature or the temperature of the meat and then it should be water activity of meat cut.

On CCP2 there really was no mention of getting rid of the drip in the other steps so please review.

When reviewing the monitoring and verification sections in the table, the authors have listed QC audit of utilized apparatuses, is this meaning calibration of equipment or consider adding calibration of thermometers and other measuring devices such as the water activity meter. Calibration is mentioned in the records section but is a key verification activity.

Line 261: The authors provide a reference that the internal temperature of frozen meat cuts must -12C maximum. It is suggested to go back and change this as the temperature of meat cuts during receiving.

Line 266: Change to microorganisms

Line 269: The reference indicates that the temperature for the thawing chamber should be lower than or equal to 4C. It is suggested that the authors change the critical limit to this instead of 4-5C. Would the authors want to suggest a water thawing method too?

Line 275: Change to italics for C. zeylanoides

Line 300: Instead of sterilized gauze should it be clean food grade gauze?

Lines 301-352: Check that the numbers and ranges are the same as discussed throughout the review.

Figure 3. Make sure to add the slice and packaging step.

Line 357: Provide information on vacuum settings and package parameters if possible. 

Line 368-373 Consider adding information about the importance of calibrating instruments for monitoring of temperatures, water activity, etc.

There are a few English issues but reviewer tried to point these out.

Author Response

Reviewer#3

Overall, the manuscript contains very important information for producing a safe and quality dry-cured pastrami. There are details that need to be taken into consideration regarding the CCPs in the HACCP plan.

Dear reviewer, our sincere thanks for taking the time to review this manuscript, and your close attention to detail. We highly appreciate your overall positive feed-back regarding the quality of the manuscript! Please see below for our responses to your comments:

Abstract: The authors have not mentioned the importance of verification in a HACCP plan. Could consider incorporating verification procedures in sentence on line 29 and 30.

Answer: Was done according to the reviewer suggestion (see the lines 27-30 of the revised version)

Lines 45-46: Prerequisite programs in quality and safety programs are the foundation of safety programs and do not result in secure and healthy food. Authors may want to consider changing to safe and quality food. Company could have prerequisite programs for a candy bar but that wouldn't be considered healthy.

Answer: Thank you for this valuable suggestion! Was done accordingly!

Lines 55-59: Change to Pastrami is a dried meat product that is curing using salt, sodium nitrite, it....for consumers.

Answer: Corrected according to the reviewer requirement.

Line 60: Consider starting as a new paragraph.

Answer: Modified according to the reviewer suggestion.

Line 61: Consider changing to E. coli O157:H7, spell out Staphylococcus

Answer: Modified according to the reviewer suggestion.

Line 63: Change to These pathogens can be traced to the raw meat and at meat manufacturing facilities on tools and machines or in drains.

Answer: Changed according to the kind suggestion of the reviewer.

Line 64: Consider changing from bad to inadequate control

Answer: Changed.

Line 67: Consider changing from dirtied to contaminated

Answer: Changed.

Line 72: Change to: The preservative use of nitrite salt...

Answer: Changed.

Line 85: Go into a little more detail to explain where the storage problems occur such as at the meat manufacture, retail store, or by the consumer.

Answer: A detailed explanation was inserted according to the reviewer suggestion. Please see the lines 87-98 of the revised version.

Line 110: Change term to phase or step of production

Answer: changed as requested.

Table 1: For Product characteristics is "absence of E. coli" mean generic or O157:H7?

Answer: Yes. Revised accordingly!

For the Storage period it looks like it was repeated in the Table. Also, do the temperatures mean the temperature of the storage room or the temperature of the product? Is there a defined or recommend shelf-life for the frozen product?

Answer: In the first column, the term “period” was replaced with “temperature” and the additional requested information were mentioned in the Table 1. Please see the lines 143-144.

Line 148: Explain what is meant by autoclave non-meat constituents? Either define the non-meat constituents or why the autoclave process. Is this something that can be done by various suppliers?

Answer: The requested explanation was inserted in the revised version. Please see the lines 161-165

Line 149: Do the authors mean generic E. coli or STEC E. coli?

Answer: Was corrected in E. coli O157:H7

Line 159: Change to city and Three percent citric acid.

Answer: Changed as requested.

Line 161: Consider changing to Fresh or raw sweet ...and hot pepper.

Answer: Changed as requested.

Line 166: Change to city

Answer: Changed as requested.

Line 175: Change to salt

Answer: Changed as requested.

Line 192: Explain approximately how many incision per an area...also how deep are the incisions and what is the diameter of the holes?

Answer: The information requesting the caracteristics of incisions were mentioned in the revised version. Please see the lines 211-212.

Line 194: Is the 6C the temperature of the refrigerator/drying chamber or the internal temperature of the meat? If temperature of room then consider adding a range such as 6C +/-1C.

Answer: The answers to the raised questions are mentioned in the lines 215-217 of the revised version.

Line 195: Is 15C temperature of the drying chamber? Again, if drying chamber temperature again provide a range or consider <15C. What was the humidity for the first drying? Consider adding washed with cold water

Answer: The „cold water” washing was added, and the humidity value was mentioned. See the lines 218-220 of the revised version.

Line 198 What was the humidity of the room?

Answer: The relative humidity of the room was lower than 75%.

Line 199: The range of 3 to 4 days is pretty wide. Consider adding in the water activity that was trying to be achieved. Also, consider including a range for 20C. Was the meat cut in the cool room prior to the drying step? If so what is the temperature of the cool room? Also what was the humidity? Did the authors measure the velocity of the fan for air renewal?

Answer: The requested information was inserted in the revised version. Please see the lines 224-226.

Line 201: If 25C for drying chamber, should it be <25C or should it be a range? What was the humidity?

Answer: The relative humidity was lower than 75%. This was mentioned in the line 227 of the revised version.

Line 203-204 Should it be cement or cemen? Check document for use of the term. Should there be a range or for 7C? Also, 4-5 days is a wide range so should the target water activity be added?

Answer: Corrected. See the lines 229-231.

Line 205: Define what is meant by excessive layer such as so many mm as for the thin layer.

Answer: 5 mm.

Figure 2. The dry curing step is for 46 hours and does not match the time of 48 hours on Line 194 and Line 194 doesn't mention dry curing on one side and then flipping and drying on the other side.

Answer: The requested information was inserted in the revised version of the figure 2, and inserted in the line 219.

For Figure 2 and Lines 209 to 211 the authors should consider a range for the 15C, 70% RH, 18C, 65 RH, 20C and 60 RH. For the 4-5 days there should be the target water activity to explain the range of 4-5 days.

Answer: The requested information was inserted in the revised version of the figure 2 and between the lines 238-244.

For the Figure 2 and Lines 192 to 211, the authors should include if there is fan control and the speed or velocity during those processing stages.

Answer: The requested information was inserted in the revised version of the figure 2 and between the lines 238-244.

Line 217: Change to up to

Answer: Changed (line 250 of the revised version).

Line 218: Change to hygiene

Answer: Changed (line 251 of the revised version).

Line 219: Change to fundamental

Answer: Changed (line 252 of the revised version).

Line 222: Change were to are

Answer: Changed (line 256 of the revised version).

Line 232: Can the authors define the velocity of medium speed?

Answer: The answer is 50m/s. See the line 266 of the revised version

Line 243: Consider changing from repeatedly segregated to repeatedly detected.

Answer: Changed (L278)

Table 3: For the Hazards make sure to spell out all of the names of the pathogens. Also, are you meaning generic E. coli or STEC E. coli? In the receiving of meat is it the internal temperature of meat that should be at -18C or the freezer temperature. In Figure 2 authors have meat cuts (-18C) leading reader to think the meat cut temperature should be -18C at receiving. Need to clarify.

Answer: Clarified in the Table 3. Please see the new version of Table 3 L290.

Table 3: Meat thawing should it be Meat thawing with room temperature at 4-5C?

Answer: Answer: Clarified in the Table 3. Please see the new version of Table 3 L290.

Table 3: The control mentions knives must be disinfected. Provide details of how this could be done such as hot water. For the meat trimming, the control indicates that meat cuts should be at 4-5C. The sentence could be improved... Well-trained workers should trim meat cuts as quick as possible to maintain the internal meat temperature stay between 4-5C.

For the Dry-curing step again it is different than in Figure 2 also address 6C and 48 hours and Relative humidity changes as discussed above. For Biological: Should it be root growth of microorganisms? For control, suggest way to obtain an aseptic knife especially in a factory setting.

For the remaining stages address the ranges or less than or greater than for the temperatures, times, RH and fan speed.

Season paste control: it would be better to indicate a clean and sanitized container and spatula are used. Sterile is not feasible in food industry.

For the Final drying is there a way to define the size of cracks or should there not be any cracks in the layer of seasoning paste. Here it is called paste is this the same as cement?

Slicing and packaging step is not included in the text after final drying or in Figure 2. Explain the parameters for vacuum packaging such as the percent of oxygen.

Answer: All of the raised concerns related by the data presented in Table 3 have been clarified in the revised version of the Table. Please see the new version of Table 3 Lines 290-291.

Table 4: Should be the first time using TVC so spell out as total viable count.

Answer: Revised in first column of the Table 4.

Again for the critical limits revisit to provide ranges or less than or greater than. This is very important in a HACCP plan especially from a monitoring and regulatory point. If the temperature in the room is only one number then it can be out of compliance if lower or higher. Also, this could increase the number of corrective actions taken.

Answer: Revised as requested, se the new version of the Table 4 (lines 402-403).

For the monitoring section would suggest including what specifically is being monitored such as the drying chamber temperature or the temperature of the meat and then it should be water activity of meat cut.

Answer: Revised as requested, se the new version of the Table 4 (lines 402-403).

On CCP2 there really was no mention of getting rid of the drip in the other steps so please review.

Answer: Revised as requested, se the new version of the Table 4 (lines 402-403).

When reviewing the monitoring and verification sections in the table, the authors have listed QC audit of utilized apparatuses, is this meaning calibration of equipment or consider adding calibration of thermometers and other measuring devices such as the water activity meter. Calibration is mentioned in the records section but is a key verification activity.

Answer: Revised as requested, se the new version of the Table 4 (lines 402-403)

Line 261: The authors provide a reference that the internal temperature of frozen meat cuts must -12C maximum. It is suggested to go back and change this as the temperature of meat cuts during receiving.

Answer: Corrected as suggested.

Line 266: Change to microorganisms

Answer: Changes as requested

Line 269: The reference indicates that the temperature for the thawing chamber should be lower than or equal to 4C. It is suggested that the authors change the critical limit to this instead of 4-5C. Would the authors want to suggest a water thawing method too?

Answer: Changes as requested and supplementary information was inserted (see the lines 303-306)

Line 275: Change to italics for C. zeylanoides

Answer: formatted in italics

Line 300: Instead of sterilized gauze should it be clean food grade gauze?

Answer: Changed according to the reviewer suggestion (line 336)

Lines 301-352: Check that the numbers and ranges are the same as discussed throughout the review.

Answer: Revised.

Figure 3. Make sure to add the slice and packaging step.

Answer: The new version of the figure was inserted in the revised manuscript.

Line 357: Provide information on vacuum settings and package parameters if possible.

Answer: 75% O2 and 25% N2 -was inserted

Line 368-373 Consider adding information about the importance of calibrating instruments for monitoring of temperatures, water activity, etc.

Answer: The requested information was inserted in the revised version (please see the lines 410-413)

There are a few English issues but reviewer tried to point these out.

Dear reviewer, once again special thanks for your tremendous review work! During the manuscript revision, each member of the research team tried to do her/his best to improve the English language. We hope that the present version has been significantly improved! In adding, the research team members believe that the remained English language issues will be solved be the MDPI professional English editing team during the manuscript proofreading after it acceptance.

Thank you again!

Dr. Imre,

On behalf of the research team

Round 2

Reviewer 2 Report (Previous Reviewer 2)

The discussion, which was previously well reduced, has now been properly adjusted.

Observe the order in which the pages appear, as Figure 3 is repeated on pages 4 and 5, but there are two numberings on pages 4 and 5. Please change.

Likewise, the flowchart is represented in different sizes on pages 8 and 9.

Author Response

Dear Editor and reviewer,

The discussion, which was previously well reduced, has now been properly adjusted.

Answer: The authors would like to thank you for your efforts reviewing our manuscript and your valuable comments helping us to improve the quality of the submission. Also, we want to thank you for your overall positive feedback regarding the quality of the previous revision!

Thank you again for your support!

Observe the order in which the pages appear, as Figure 3 is repeated on pages 4 and 5, but there are two numberings on pages 4 and 5. Please change.

Answer: with respect to your request, the corresponding author is not able to solve the continuous page numbering concern, even if in the manuscript document the total number of pages (n=29) are well indicated, but the current page is wrong starting with the page no. 11. However, the research team hope that this issue will be solved by the professional MDPI team during proofreading!

Thank you for your understanding and cooperation!

Likewise, the flowchart is represented in different sizes on pages 8 and 9.

Answer: the raised concern was solved!

This manuscript is a resubmission of an earlier submission. The following is a list of the peer review reports and author responses from that submission.

Round 1

Reviewer 1 Report

Perceptions and suggestions:

Page 1, Abstract, line 19-20 “so the present investigation is demeed to be as a completion for these.”: This part is confusing, adjust to make it clearer.

Page 1, Abstract, line 27-28 “guarantee getting product of zero defects.” rephrase this point, as food products should not only "guarantee getting product of zero defects": in the first moment of their process, before they should guarantee a safe food.before should guarantee a safe food.

The introduction does not clearly present what dry-cured pastrami is, it does not clearly insert the problem that justifies the importance of this article.

Page 2, Introduction, line 70: “GMP in animal fabrication”:

Page 2, Introduction, line 71: “Hereafter, the conclusions were exploited to stablish a HACCP strategy for all anticipated”, where the expectation was to contain the clear and direct objectives of the article, this part that I did not understand is presented.

Page 4, Table 1, “Longissimus dorsi muscle”: ajust for Longissimus thoracis et lumborum (Meat Sci 1990;28(3):259-65. Recommended terminology for the muscle commonly designated “longissimus dorsi”. Doi: 10.1016/0309-1740(90)90010-4).

Many points in the text seem to present problems in the writing of the English version.

The results and discussion are not presented as a normal scientific article. The information is all in the materials and methods. The objectives and hypotheses are not clear and consequently the conclusions reflect this.

This material looks much more like a manual of good practices than an article.

Author Response

Reviewer #1

Dear reviewer,

Thank you so much for your review, kind comments, and valuable suggestions. We have modified the text according to them.

Page 1, Abstract, line 19-20 “so the present investigation is deemed to be as a completion for these.”: This part is confusing, adjust to make it clearer.

Changed in: “The current scientific paper is considered as a completion of the two previous publications.”

Page 1, Abstract, line 27-28 “guarantee getting product of zero defects.” rephrase this point, as food products should not only "guarantee getting product of zero defects": in the first moment of their process, before they should guarantee a safe food. before should guarantee a safe food.

Changed in: “… to assure producing safe food without defects”

The introduction does not clearly present what dry-cured pastrami is, it does not clearly insert the problem that justifies the importance of this article.

In the line 51 the following paragraph was inserted:

“Pastrami is a dried meat product which cured using salt, it goes through several manufacturing stages under special conditions of temperature and relative humidity and it may be a hazard for consumer. Since the Egyptian pastrami is one of popular and desired meat products among the Egyptians; the researchers and industry officials had to set rules that ensure the safety of pastrami.”

Page 2, Introduction, line 70: “GMP in animal fabrication”:

Changed in “GMP in animal rearing and meat manufacture.”

Page 2, Introduction, line 71: “Hereafter, the conclusions were exploited to stablish a HACCP strategy for all anticipated”, where the expectation was to contain the clear and direct objectives of the article, this part that I did not understand is presented.

This part is referring to conclusions related to previous researches about veterinary drugs, preservatives, allergic additives, animal feed containing mycotoxins and heavy metals. All these previous matters are considered at HACCP establishing. We can adjust it in line No. 71.

Page 4, Table 1, “Longissimus dorsi muscle”: ajust for Longissimus thoracis et lumborum (Meat Sci 1990;28(3):259-65. Recommended terminology for the muscle commonly designated “longissimus dorsi”. Doi: 10.1016/0309-1740(90)90010-4).

The amendment is applied in Table 1 according to your suggestion.

Many points in the text seem to present problems in the writing of the English version.

The authors acknowledge the fact that the English content of the manuscript need some improvements. According to the reviewer suggestion, during the manuscript revision, a native English speaker colleague helped the research team to improve the English content of the manuscript. In addition, the authors consider that the improvement of the English language is solvable during the final “Pending English” status before publication, when the paper will be edited and finalized by the MDPI team.

Thank you for your understanding and consideration!

The results and discussion are not presented as a normal scientific article. The information is all in the materials and methods. The objectives and hypotheses are not clear and consequently the conclusions reflect this.

This material looks much more like a manual of good practices than an article.

  • The current research in its entirety is applied research and it is a form of systematic investigation that involves the practical application of science, adopting and using parts of research, accumulated theories, knowledge, methods and techniques, and the purpose of research is to serve the state or business.
  • Many research papers in the field of dry-cured meat dealt with many important points in the field of manufacturing. The research team also discussed the study of pastrami manufacturing as a dry and cured product through two previous studies as follows:
  • The first research paper discussed the effect of pastrami storage temperature and period on the microbiological quality of pastrami, its sensory and physico-chemical properties.
  • The second paper concluded that although the beneficial effect of non-meat ingredients used in pastrami processing and manufacturing, those ingredients led to an increase in the numbers of microbes and mycotoxins. Also, the autoclave sterilization of non-meat ingredients and the use of HACCP-applied spices led to a reduction in the numbers of one of the most famous and important food-borne microbes, Escherichia coli, and it also led to a reduction in aflatoxins.
  • The research team found that it is necessary to collect all the conclusions related to pastrami and dry-cured products in order to establish a tight HACCP system for one of the most important meat products not only in Egypt but also in many countries of the world.
  • I do not think that the issue of manufacturing Egyptian pastrami or applying the HACCP program to it is a local issue related to Egypt only. There are many manufacturers in developing countries who want to know new products, methods of safe production and introduce such products to consumers in their countries.
  • Therefore, the idea of this research came to set up a HACCP system for pastrami, where the HACCP is considered a protective fence from many economic risks related to the loss of a polluted product, which incurs large losses to manufacturers.

Thank you again for your time and suggestions!

Reviewer 2 Report

This is an article with the use of HACCP in a meat product typical of the region. In general, there is no relevant scientific contribution because it is a well-known topic. There are text constructions with short and segmented sentences, making the reading a bit fragmented. It presents repeated texts with information already described in the flowchart, such as Figure 2. It must be revised.

Author Response

Reviewer #2

This is an article with the use of HACCP in a meat product typical of the region. In general, there is no relevant scientific contribution because it is a well-known topic. There are text constructions with short and segmented sentences, making the reading a bit fragmented. It presents repeated texts with information already described in the flowchart, such as Figure 2. It must be revised.

Dear reviewer,

Thank you so much for your review, kind comments, and valuable suggestions. We have modified the text according to them.

Our answers to the raised concerns are the following:

  • HACCP is already well-known as a system and principles. We are a research team that does not re-establish the well-known principles of HACCP, but apply them to a specific product according to the results obtained before. So, the paper is considered as applied research investigation that involves the practical application of science, adopting and using parts of research, accumulated theories.
  • HACCP as a program is well known, but its application to a specific product may cause a confusion among many manufacturers and even quality officials because each product has its own specificity and manufacturing conditions.
  • It is true that a lot of papers has dealt with manufacturing conditions, temperatures, and pastrami properties, but in this scientific paper we have combined these results to form a scientific approach through a very important preventive system for pastrami product according to its specificity.
  • Not every manufacturer or quality official will be able to collect scientific papers related to the development of each product manufacture and draw their results to develop a HACCP plan for the product on a scientific base, so that was the role of scientific research.
  • Therefore, in this research, we relied on gathering the scientific results and conclusions into a realistic practical application, as many manufacturers find it difficult to identify the critical points during the various stages of manufacturing. Therefore, they can be guided by the critical points contained in this research paper, which are based on scientific results.
  • The text constructions have been revised.
  • With respect to the reviewer opinion the authors would like to maintain the data presentation in a combined form between the text and the presented flow diagram of the dry-cured pastrami production

Reviewer 3 Report

The manuscript is presenting and described particular stages establish HACCP tactic for hazards related to Egyptian dry-cured pastrami production.

In the reviewer's opinion, the manuscript submitted for review does not meet the requirements for scientific papers. This can be seen in the structure of the manuscript, which not the necessary chapters for scientific papers, i.e.: results and discussion. There is also no appropriate manner described research methodics. In fact, the paper is a description of the procedure introduction ensuring food and consumer health safety, dedicated to a specific industrial plant. An important drawback of the manuscript is the lack of clearly and legibly indicated the aim of study.

The Authors indicate some of technological operations (for example line 130-131) but do not provide the process parameters. Namely, what on the reason did the authors stated that: "The utilization of HACCP treated high-quality and autoclaved non-meat constituents enhanced bacteriological quality through minimizing microbial content (total bacterial count and E. coli) and affirmed manufactured pastrami safeness via destroying E. coli O157:H7”? – vide line 130-132. Has the determined the effect of the modified formula of the curing mix on its antimicrobial impact on the final product? If so, by what method/methods? Next, Line: 179-182, what was the weight of the piston/pressure expressed in N/meat surface unit?

Nevertheless, the indicated shortcomings, the content of the reviewed manuscript is interesting and valuable, but in the reviewer's opinion the manuscript content should be corrected.

In my view English style should be proofed by native speaker – reviewer's recommendation.

Author Response

Reviewer #3

In the reviewer's opinion, the manuscript submitted for review does not meet the requirements for scientific papers. This can be seen in the structure of the manuscript, which not the necessary chapters for scientific papers, i.e.: results and discussion. There is also no appropriate manner described research methods. In fact, the paper is a description of the procedure introduction ensuring food and consumer health safety, dedicated to a specific industrial plant. An important drawback of the manuscript is the lack of clearly and legibly indicated the aim of study.

Dear reviewer,

Thank you so much for your review, kind comments, and valuable suggestions. We have modified the text according to them.

Our answers to the raised concerns are the following:

  • The current research in its entirety is applied research and it is a form of systematic investigation that involves the practical application of science, adopting and using parts of research, accumulated theories, knowledge, methods and techniques, and the purpose of research is to serve the state or business.
  • HACCP is already well-known as a system and principles. We are a research team applies such principles to a specific product according to the results obtained by other researchers. So, the paper is considered as applied research investigation that involves the practical application of science, adopting and using parts of research, accumulated theories.
  • The research concluded critical control points that can be found along the processing procedures which represent a confusion for many producers and even quality engineers who do not know all the manufacturing updates related to the dry-cured pastrami.
  • The dedication of specific industrial plan in the current paper came to exploit the applied approach to apply a scientific approach like HACCP in pastrami manufacture for the benefit of manufacturers.

The Authors indicate some of technological operations (for example line 130-131) but do not provide the process parameters. Namely, what on the reason did the authors stated that: "The utilization of HACCP treated high-quality and autoclaved non-meat constituents enhanced bacteriological quality through minimizing microbial content (total bacterial count and E. coli) and affirmed manufactured pastrami safeness via destroying E. coli O157:H7”? – vide line 130-132. Has the determined the effect of the modified formula of the curing mix on its antimicrobial impact on the final product? If so, by what method/methods? Next, Line: 179-182, what was the weight of the piston/pressure expressed in N/meat surface unit?

For line 130-131, it was a clarification for HACCP application importance in dealing with non-meat constituents added in pastrami manufacture. This was a conclusion for a previous scientific paper implemented by authors and mentioned in references as follows:

Abd-Elghany, S.M.; El-Makhzangy, A.M.; El-Shawaf, A.-G.M.; El-Mougy, R.M.; Sallam, K.I. Improving safety and quality of Egyptian pastrami through alteration of its microbial community. Lwt 2020, 118, 108872. https://doi.org/10.1016/j.lwt.2019.108872

The authors found that although the beneficial effect of non-meat ingredients used in pastrami processing and manufacturing, those ingredients led to an increase in the numbers of microbes and mycotoxins. Also, the autoclave sterilization of non-meat ingredients and the use of HACCP-applied spices led to a reduction in the numbers of one of the most famous and important food-borne microbes, Escherichia coli, and it also led to a reduction in aflatoxins.

Therefore, it was necessary to ensure the safety of all components involved in the manufacturing process, even the curing mixture or seasoning paste, because it may lead to an increase in the microbial count or mycotoxin, according to the research paper previously referred to, which was carried out by some members of the research team before.

The pressing power of piston was mentioned as required.

Nevertheless, the indicated shortcomings, the content of the reviewed manuscript is interesting and valuable, but in the reviewer's opinion the manuscript content should be corrected.

Comments on the Quality of English Language

In my view English style should be proofed by native speaker – reviewer's recommendation.

The authors acknowledge the fact that the English content of the manuscript need some improvements. According to the reviewer suggestion, during the manuscript revision, a native English speaker colleague helped the research team to improve the English content of the manuscript. In addition, the authors consider that the improvement of the English language is solvable during the final “Pending English” status before publication, when the paper will be edited and finalized by the MDPI team.

Thank you for your understanding and consideration!

Round 2

Reviewer 3 Report

In the reviewer's opinion, the changes made to the manuscript are minor and, in principle, it does not differ from the original version. The structure of the manuscript is still not appropriate for scientific papers – only the chapter "Materials and Methods" has been separated. There are still no "Results" and "Discussion" sections in this paper.

After a careful second review of the manuscript's scientific merit, I believe that the present paper was don't corrected according to the reviewer's recommendations. Therefore, I cannot recommend this paper in present version to published in Foods scientific journal.